# An Extensive Preliminary Blockchain Survey from a Maritime Perspective †

**Rim Abdallah** [1,2,*], **Jérôme Besancenot** [1], **Cyrille Bertelle** [2], **Claude Duvallet** [2] and **Frédéric Gilletta** [1]

1   HAROPA PORT, 76600 Le Havre, France
2   LITIS, The Laboratory of Computer Science, Information Processing and Systems, University of Le Havre Normandy, 76600 Le Havre, France
*   Correspondence: rim.abdallah@etu.univ-lehavre.fr
†   This paper is an extended version of our paper published in 4th Congress on Blockchain and Applications.

**Abstract:** The maritime industry is moving towards a digital ecosystem to achieve substantial mutual profits. To achieve this, there have been attempts to combine existing, disjointed systems into more efficient, standardized platforms that can be scaled up. However, this transition has faced challenges. To address these issues, it is suggested that innovative technologies such as blockchain be utilized due to their alignment with the sector's needs. This study uses a triangulation approach by examining a mix of literature, web-based data, applications, and projects to showcase the contribution of blockchain and its potential use cases. We also explore its potential use cases based on other sectors using projection and parallelism. Additionally, the study delves into limitations and possible solutions. This research acts as a preliminary study for the implementation of blockchain in the maritime industry, and advocates for its use as a revolutionary approach. The findings will be beneficial for scholars, policy makers, and practitioners in the maritime industry.

**Keywords:** blockchain; smart port; shipping industry; maritime sector; decentralized technologies; connected ports





## 1. Introduction

For decades, shipping has been at the forefront of global economic growth and is now responsible for transporting 90% of trade goods. Maritime shipping is deemed the safest mean of transporting cargo across countries, in addition to being cost effective for international trade. This is why the industry embodies a pole of attraction for many enterprises responsible for shipping and port-related activities. These organizations evolved from adjacent and agglomerated business exchanges in port zones to business clusters. As shipping methods advanced at a rapid pace, the demand for global goods increased, leading to the mass transportation of containerized freight via maritime routes. The rise of containerized freight created the need for a hub to serve as a mediator among the cluster members. The hub would ensure that high-capacity and frequent services were provided to keep up with the complexity of mass transportation, which differs from the previous method of bulk shipping where individual materials were shipped. Ports served as the central hubs for these cluster nodes to maximize economic benefits across a globally spread shipping market.

Consequently, the massification, need for an intermediary, and port clustering have led to the formation of more evolved topologies such as port community systems (PCSs). PCSs were introduced as a unifying platform to facilitate business exchanges amongst the cluster's stakeholders, fulfilling standardization needs. The first PCSs, established in European ports, can be traced back to the late 1970s.

PCS's benefits were rapidly proven as new PCS formation spread across the globe, making the supply chain a series of PCS nodes. PCS was depicted as a good governance

and efficient tool per the UNCEFACT's Recommendation No. 33 with benefits for both governments and businesses. It shifted exchanges in those clusters to a single window (SW) concept. Previously, the same documentation would need to be presented multiple times to distinct agencies with their own systems, whether physical or digitized [1]. With the SW concept, however, all necessary documentation could be presented through a single electronic interface. SW required only a single submission of relevant documentation at the PCS. The PCS is responsible for the informational exchange amongst its stakeholders, as depicted in Figure 1 below.

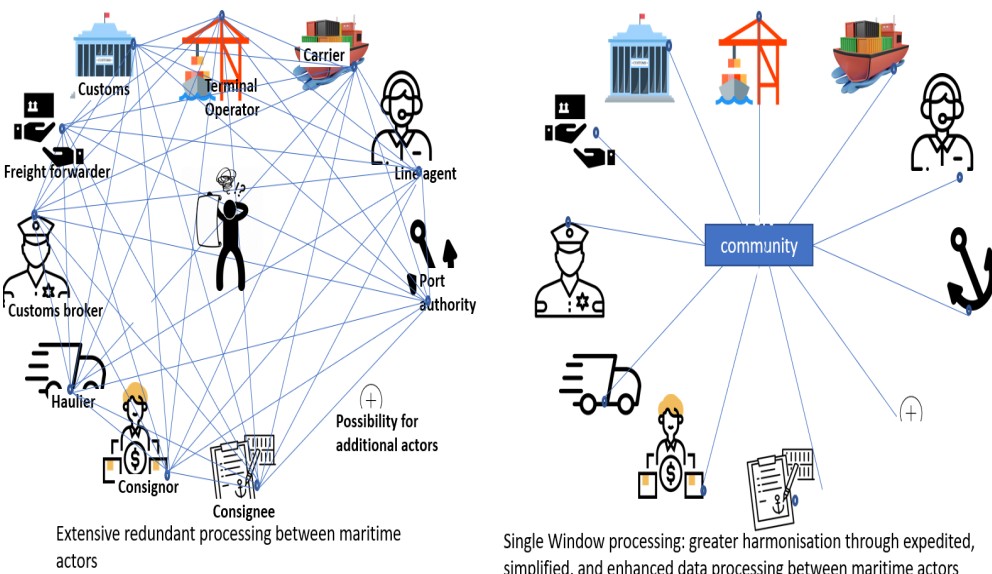

**Figure 1.** An exemplary representation of port community members before and after PCS: a single-window solution elaborated by the author.

PCS lived up to its definition by the International Port Community System Association (IPCSA). It was defined as a neutral, open digital platform that enables safe information sharing amongst port-related stakeholders to ensure ports' greater competitive advantages [2]. The benefits of PCS extended beyond efficient resource allocation and fee collection by governmental bodies. It also resulted in better business compliance, improved security, and a decrease in fraudulent activities. Moreover, it provided several business benefits, including reduced fares, speedy processing, standardized and more comprehensible procedures, clearer rules, and increased transparency. In addition to contributing towards more sustainable logistics and carbon reduction [3].

In general, PCS streamlined data exchange and trading processes, simplified alignment with international standards, and set the founding stone for process automation. It is difficult to generalize the exact functionalities of PCS because they vary depending on the community's actors and their relevant metrics [4]. Furthermore, governance models differ from one PCS to another (public service, tool, landlord, and private port) [5]. Actors' roles within a PCS eventually advocate the use of technological advancement for business maintenance informational, capital, and logistical exchange and delays limitation. An example of typical PCS actors is shown in Figure 1: port authorities, ocean carriers, customs, etc.

*1.1. A Cumbersome Global Digitization*

PCSs are considered a setting stone in the digitization roadmap. Single-window systems are favored on a global scale and can federate PCSs integration. Nevertheless, each PCS across the supply chain has its distinct network participants, not necessarily including all relevant parties. Furthermore, the work requirements of establishing a single-window system mentioned in [6] can be divided into two main categories: negotiations and technical work. This aligns with the observation of the IPCSA, that not only do actors differ

from one PCS to another but so can IT infrastructure and functionality. The analysis of the tasks and attribution to these categories clearly shows that the majority of the work is negotiating, as shown in Figure 2, which can mostly be linked to the distinct information systems of each participant. Although standardizing documents is a crucial step in PCS implementation, inconsistencies exist among different PCSs due to varying governance models and technological infrastructures. Consequently, integrating different PCSs with each other can be very challenging. Therefore, PCS focused on creating a beneficial centralized data exchange platform without taking into consideration the network's architectural aspects and scalability. Each PCS has indeed offered the promised local economical gain and efficiency. However, the global scale remained untouched, and the network remained fragmented into modular centralized systems. As a result, redundancy persisted, and automation possibilities remained limited. The lack of communication between different PCSs is partially due to the appearance of PCS before the advent of peer-to-peer networks which allowed the establishment of solutions on a global scale. Indeed, it has hindered the globalization process, but can be the challenge that can be extended beyond distinct technical infrastructure to the different governance models. PCSs are considered key competitiveness factors, and open data exchange is perceived as detrimental to their gains. Additionally, not all pertinent stakeholders are always included in a PCS network, further exacerbating the issue.

Even with global efforts made by the IPCSA and the international maritime organization (IMO) that encourage port-to-port collaboration and data exchanges, these efforts extend only to recommendations that cannot supersede the rivalries present, the different governance, and PCS ownership models of whom without their consent data exchange remains impossible.

## Single window strategic requirements

**Figure 2.** Actors' strategic requirements analysis for a single-window system elaborated by the author.

While PCSs can offer great competitive advantages and beneficial gains on a local scale, fragmentation and the lack of reliable data on a global scale can hinder shipping processes for cargo [7]. Moreover, data abundance in centralized systems circulating amongst heterogeneous stakeholders can present intelligence risks. Although PCSs have many advantages, skepticism prevailed, which has resulted in a reluctance to share transparent data. This has created ambiguity and an obstructive environment that leads to idle scrutiny, particularly for governmental units where precision is crucial for security reasons.

### 1.2. Change Prospects vs. Innovational Resiliency

Consequently, a behavioral change started appearing in the maritime sector and sustainability strategies. For example, the internationally widespread organization Maersk

started searching for more effective digital solutions. These solutions needed to overcome global inter-organizational obstacles in the supply chain, such as the multitude of actors and the complexity of regulations, while also ensuring a minimum of efficiency and digitization like the previously deployed intra-organizational solutions [8]. In addition, any revolutionary solution should be lucrative enough to generate sufficient interest from other maritime actors to revolutionize the industry. This was confirmed by Knut Ørbeck-Nilssen, the Chief Operating Officer and President of DNV GL and Maritime and Director of Division Europe, Africa and Americas, in 2017. He emphasized that any technological innovation should also promote collaborative approaches to succeed, because no maritime actor can act alone facing the opportunities to reduce costs and facilitate exchanges [9]. Furthermore, the introduction of new technologies to the maritime ecosystem required new technical capabilities which meant introducing additional costs in a gain-driven environment. This is why being lucrative was a key challenge in addition to a successful implementation with an infrastructure that was not designed for global digitization, and where data is vigilantly protected and very stingily shared. An optimal evolution of existing systems, in theory, grants global real-time access to shared data amongst the maritime network of participants forming a unified and trusted source of information and transforming the fragmented systems of the supply chain into a unique more evolved collectively maintained solution. This became technologically more achievable with the new and emerging technology called "Blockchain".

The alignment between blockchain concepts and the omnipresent digitalization needs of the maritime ecosystem is noteworthy. These needs can be portrayed as the optimum digitized evolution for a sector with a multideity of participants. The technology has increasingly captured interest across various sectors. It became most reputable for its application in the financial sector, namely, "bitcoin". Initially, it was introduced as an alternative to the digital currency that is double-spend resilient. Since digital currencies are virtual concepts, they became much easier to duplicate than regular physical currencies, despite their advantageous qualities to facilitate and simplify exchanges. The action of paying more than once with digital currency is called a "double spend". Blockchain stacked previously existing technologies in a new and innovative approach to prohibit double spending [10]. An entity under the name Satoshi Nakamoto suggested the use of timestamps and usage of the cryptographical algorithm [11] to certify the date of creation and or modification of digital data and to protect the privacy and digitally sign data, respectively. Furthermore, a public database was suggested to be distributed to a pool of users that kept track of the electronic register whose pages consisted of a series of transactions grouped into a block, and each block was linked to its previous incorporating its resulting hash, as displayed in Figure 3.

### 1.3. Research Rational

Blockchain technology is the upcoming quantum leap for the maritime industry as its characteristics coherently parallel technological literary recommendations for revolutionizing the industry [1]. It facilitates the transition from globally fragmented centralized systems to a peer-to-peer network without the need to establish complete trust between actors. Real-time data accessibility is granted through the distributed append-only digital registry that is collectively maintained through a consensus mechanism. The consensus mechanism is the rule by which the ledger is extended, and additional data is recorded that guarantees nodes' honesty. An example of consensus is the PoW (proof of work), as in other consensuses it demands a certain promise to be made from the node to the network. In the case of PoW, the node's computational power is presented as a fee to find the network's difficulty level (nonce) that would allow the addition of the block to the ledger and would also be incorporated in the ledger's hashing chain. The timestamping, encryption, and chaining of the data establish immutability that consequently elevates tracking, tracing, and certifiability for the sector. Additionally, several advantages result from the use of blockchains such as the alleviation of cyber burdens and data protection through cryp-

tographic algorithms and encryption, in addition to the elimination of single points of failure and the need for trusted intermediaries existing in centralized systems through decentralizations and automation via smart contracts. Hence, this technological shift might engender a colossal effect on the sector with an unprecedented estimated 15% improvement and growth in world trade [12] and reduction of transportation costs by 20% [13]. Likewise, digitization and smart contracts provide increased personalized automated services. Moreover, Ref. [8] highlights an achievable 40% decrease in delivery delays.

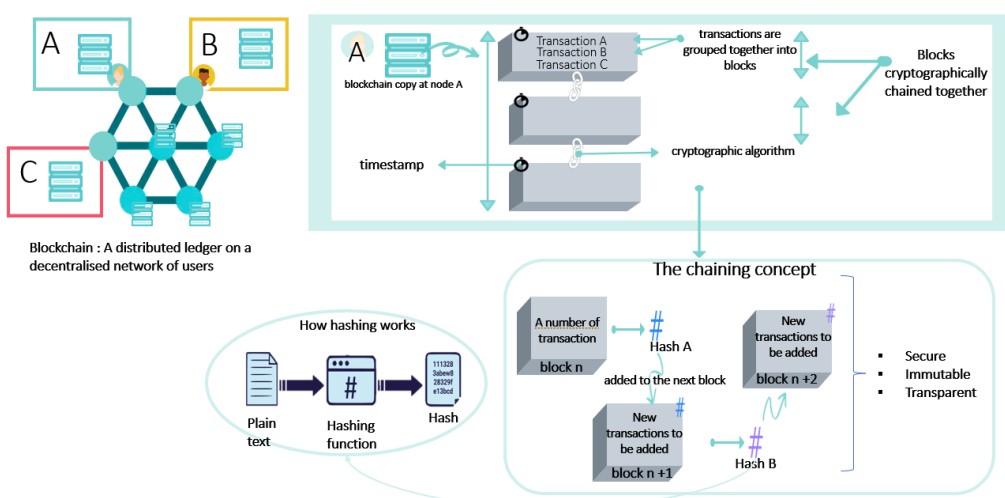

**Figure 3.** Blockchain concepts elaboration by the author.

However, the sector is usually regarded as technologically prudent and conventional [14,15]. This hypothesis is pinpointed not only in reports where the shipping industry is classified as having one of the smallest innovational impacts [16] but also appears in the literature [17]. It was demonstrated through the maritime literature that advancements in the shipping industry have always been met with a certain level of conservatism in comparison to other industries, whether on the technological or the technical level [15]. Moreover, as demonstrated in [14] most of the studies focus on the assessment and effects of the adoption of a particular innovation in a narrower sector: the port, and most specifically, terminal operators. Despite it being a very essential player in the supply chain industry, a successful innovation path should incorporate broader coverage to answer to the totality of intertwined businesses that fabricate the maritime ecosystem.

Hence, this technological delay is not due to the lack of motivation by the sector nor by any limited technological capabilities, but by a lacuna between innovative approaches and attempts and achievability [14,18]. In the gain-driven maritime industry, innovation is often restricted by profit-driven considerations, as defined by [18]. According to their definition, innovation is the effort to reduce costs and increase gains through a revolutionizing, sector-altering approach. For the most part, despite partial acknowledgment of the technology's potential, groundwork in the sector investigating technology is disintegrated into specific cases [18,19]. Paradoxically, a global amassing approach continues to miss halting blockchain revolutionizing effects.

Therefore, eventually providing a theoretical overall global approach, grouping fragmented case studies hoping to revolutionize the processes of the industry, while also maximizing revenues and maintaining healthy competition between maritime actors, is necessary. The aim of this paper is to provide a comprehensive review of blockchain technology in the maritime sector, filling gaps in the existing literature. Through a systematic review, assessment, and examination of the potential and challenges of this technology in the sector, this paper aims to establish a foundation that aligns the technological characteristics of blockchain with the industry's needs. This paper also seeks to present a theoretical approach that brings together fragmented case studies and offers the potential

to revolutionize the industry while also maximizing revenues and maintaining healthy competition between maritime actors. This is done through a triangulation approach [20], crossbreeding web-based, extensive literary research and technological advances. We aim to accelerate blockchain endorsement by the shipping industry overcoming the nescience of the wide spectrum of opportunities offered by the technology. This will allow us, finally, to unleash the actual theoretically revolutionary results that can be achieved while being fully aware of the obstacles and challenges and evading routes for a smoother deployment.

## 2. Research Methodology

For a more enriching contribution to the maritime sector, our research was based on a triangulation approach, a method broadly detailed in the literature [21,22]. This method is used by researchers to conceive a better understanding of a phenomenon using multiple investigatory approaches, since one is not quite enough. Consequently, prelusive hypotheses are set and then corroborated by other approaches for more enlightening insights. The first investigation phase was an extensive systematic literature review and a state-of-the-art process for the technology itself, the maritime sector, and innovation and technology within it. A specific process establishes a broader reflection on these topics and a qualitative analysis of the present sector's position and patterns through the literature [23,24]. The main aim was also to group the fragmented literature into a whole extensive study, creating a body of literature focused on blockchain in the maritime ecosystem. Afterwards, web-based research was performed on different companies' and solutions' websites. This allowed us to cross-match the previous work with the eventuality and actual development activities and reflect on further hypotheses concerning barriers to the implementation of the technology. Finally, the two previous steps were repeated, but instead of blockchain implementation, we scoped challenges that were deducted from the previous step, and consequently proposed possible solutions and analysis.

## 3. Blockchain: A Potential Contributor for Revolutionary Maritime Evolution

The attempts to establish new evolutionary approaches in the maritime sector retained findings and objectives set by the "industry 4.0" concepts. They aimed to achieve global interoperability by utilizing innovative technologies while maintaining a competitive edge for businesses. This was also evidenced in the new shipping business model which revolved around value.

### 3.1. A Profitable Argument

Value is created when the profit exceeds expenses and expands beyond tangible assets. It can consequently be generated through indirect revenues such as new technologies and scalable networks [25] embracing a more heterogeneous landscape of systems. An example of this can be portrayed through the shipping cost equation which accelerated the shift from bulk shipping to containerized cargo. Profit (P) can be calculated by subtracting expenses (C) from revenues over a period of time (t) as shown in the following equation:

$$R(t) - C(t) = P(t) \tag{1}$$

The cost of a cargo (C) is equal to the sum of all costs (operational cost, cargo maintenance, cargo voyage, cargo handling, and cargo capital costs). The operational cost also covers the vessel's operational cost. The bigger the vessel, the bigger its operational cost. However, C is inversely proportional to the vessel's size and also inversely proportional to the overall cargo size. This means that despite that the bigger the vessel the larger its operating costs, these costs are outweighed by the revenue increase for cargo, thus lowering its costs and maximizing profits in the above equation. We base our reasoning on this and the logic in [26] which led to the deduction that added control over operational parameters can also minimize costs. An example of reasoning can be that the non-tangible asset, the ship's idling time, where added control reduces error by a third, largely improves statistical analysis and consequently performance to minimize costs [26]. Moreover, a ship's revenue

(R) is directly linked to its productivity factors such as operational planning, backhauls, operating speed, off-hire time, dead-weight utilization, and port time. These factors can also be considered as operational parameters where added control also increases revenue, creating a larger profit. In an ecosystem that is largely driven by profit, having presented the importance of added control such as transparency and traceability for optimal operational parameters and profit increase, we argue that blockchain can be introduced as a revolutionary approach to the maritime 4.0 ecosystem for an increased value [2].

### 3.2. About Blockchain

Blockchain technology enables the distribution of infrastructure across both inter- and intra-organizational entities, without the need for a centralized authority that is trusted. The technological advancement in the shipping industry is currently embodied by the use of PCS and data-driven tools. However, operational efficacy and logistics management are still far from optimum across the supply chain, with poor global transparency and data exchange. Moreover, the supply chain remains haunted by the aberrant use of physical documentation halting any-real time access to information and decreasing overall efficiency and accountability. Despite the introduction of advanced and increasingly digitized systems in fragments of the supply chain, as a whole, it is still fragmented. It consists of successively intertwined processes and actors having their own distinct systems, where any detrimental variable can cascade over the whole chain as shown in Figure 4.

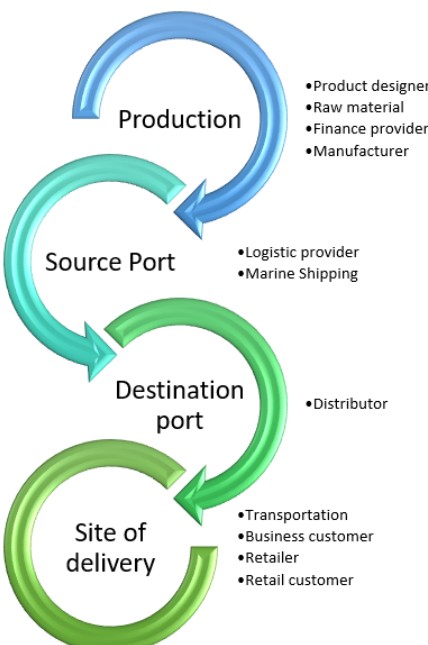

**Figure 4.** A simplified scheme of the supply chain elaborated by the author.

With blockchain, the information register can be replicated for all relevant actors beyond any predefined local networks such as PCS or cargo community systems. It allows real-time dissemination through all needed checkpoints over the supply chain. It creates a decentralized distributed communication system where data can be verified and certified through consensus protocols. This aligns with the distinctive results presented in [27] that highlight the importance of decentralized approaches and unconventional representation of the supply chain beyond port zones. The maintenance of such systems does not rely on any unique centralized trusted entity but instead on the infrastructure of the system itself. Decision making, processing, auditing, maintenance, transaction approvals, and data validation is carried out collectively. Moreover, actors are forced to maintain a minimum of reliability for not only the collective convenience but only their own individual gains. Any inconsistency can be quickly detected and unvalidated through the replicated ledger,

data chaining, time stamps, encryption, and the collective consensus protocol. Such a system unlocks further transparency and traceability that consequently reduces fees including accounting and auditing, unnecessary trusted intermediaries and single point of failures [28].

Nevertheless, using blockchain is not analogous to data divulgence. As mentioned, blockchain technology was deployed as an infrastructure for digital currency. It established trust within a trustless environment based on a peer-to-peer system, and allowed users to exchange financial transactions without trusted centralized establishments (such as banks) while also freeing currency concepts from being entangled with countries' economical statuses and banks. This explains the need for such an application to be public, as it is deployed within a public pool of users, where various unrelated actors are responsible for ledger examination, maintenance, and data validation, declaring it as a valid trusted platform used as a global digital currency-exchange liaison. However, the financial sector is quite different from the maritime sector. In the latter, despite the need for distributed global approaches, the presence of centralized entities such as governments remains necessary, and is a mandatory consideration for any technological deployment, in addition to the presence of sensitive data that comes along with all the competitive entities that negate the concept of an open data-sharing platform for competitiveness and security reasons, among others. This brings us to reflect on other types of blockchain. such as private or consortium, where we have more control over the blockchain network and data. The technology can be introduced to the shipping industry by the creation of a hybrid network that includes clusters of public, private, and consortium networks communicating among each other [28–30]. It is also important to distinguish between each type's characteristics for it to answer to the maritime ecosystem's constraints. For example, an end user or a retail customer can be a part of the public globally spread blockchain network communicating with other types of networks that consist of shipping agencies, ports, freight forwarders, and transportation agencies. It would grant the user transparent and clear visibility of its cargo's journey (such as arrival time) while maintaining accountability at its optimum without the need to divulge sensitive irrelevant data, such as the ship's manifest. A theoretical conceptual representation of the envisioned system without a pre-defined set of network types is represented in Figure 5.

The network choice should not be only based on literary information and each type's characteristics, but also should include maritime actors' experiences through surveys, and studies to reach effective optimality between decentralization, trust, and immutability. In a public blockchain, trust is established through decentralization and immutability. The replicated ledger over a larger number of nodes makes it harder to alter information in the ledger since each block is linked to the previous one.Therefore, to alter one line of information, we would need to first recalculate one whole ledger to make sure the alteration is included in the chaining process. Then we would also need to disseminate the newly altered ledger over all or the majority of the nodes present in the network. The larger the network and bigger the ledger, the more time- and computationally-consuming it is, which renders public networks immutable. In private or permissioned networks, the number of nodes and ledger size is more limited, making them less immutable.

### 3.3. Partial Recognition and Early Adopters

Blockchain has successfully evolved in the financial sector with several successful projects such as bitcoin, Ethereum, and others that have been adopted on a global scale. A user in a time zone A with a native currency X can transfer a bitcoin to another time zone B with a native currency Y independently of any singular trusted entity or bank, relying only on the Bitcoin network. After a certain time, the transaction can be successfully observed and is certified and verifiable. This seamless exchange is not only restricted to financial assets but expands to include tangible and intangible assets. This conforms the hypothesis that blockchain can be used in the maritime sector for more seamless operational processing and digitization. In fact, it has been reflected as the technology has successfully

sparked interest in other sectors, including the maritime industry, where multiple projects emerged (see Table 1 [1]).

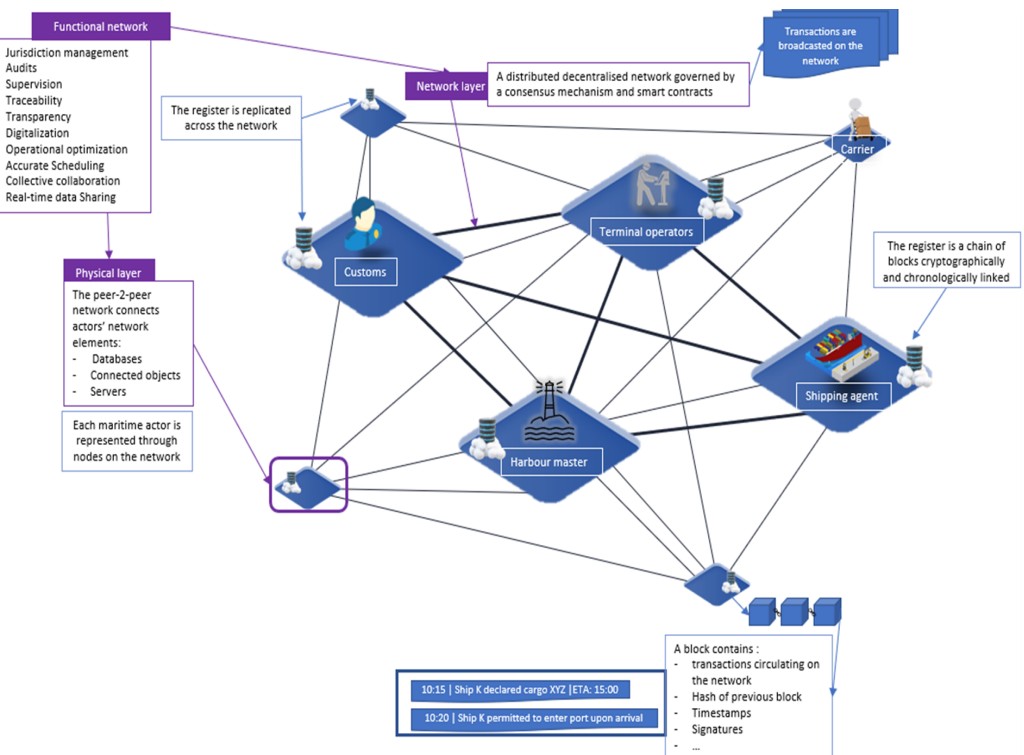

**Figure 5.** A conceptual blockchain-based maritime ecosystem elaborated by the author.

However, not all projects were successful (by successful, we restrict the interpretation of the word to implying sustainability or survival). This is because we cannot judge the idea behind the project nor its implementation. as some were correctly implemented but failed to survive in the ecosystem. These projects differ in concept and cover multiple areas of the logistical maritime operations, from storage to delivery and payments, as well as the improved transparency and security achieved. For example, the project "TradeLens" is a joint idea between two competent companies, IBM and Maersk, with an ultimate goal of cost-effective improved traceability. Another example is "CargoX" which focuses on document digitization, such as bills of lading. It aims to provide a secure, reliable, and time- and cost-effective method of processing shipping paperwork anywhere on the globe. Other projects went further and targeted payment methods in the maritime industry with an idea of a global decentralized currency such as ShipChain or 300Cubits. ShipChain created a SHIP token to facilitate transactions. However, this project failed to overcome the regulatory novelty of the technology. The initial coin offering that put the token into circulation was unregistered with the security and exchange commission. The project shut down after a huge payment to settle its charges and cut down its resources. The 300Cubits project idea was to create TEU tokens targeting the problem of no-shows and rolled cargo. Similarly, the project faced certain challenges that obstructed its adoption. This could be due to the lack of inherent interest in the project's concept by the maritime ecosystem. The project over-complicated the problem. Cargo overbooking and rolling may be solved with a simple edited freight contract. Additionally, there is no currently pressing change to induce a contractual change; nevertheless an investment in a new technological solution is required.This state-of-the-art work highlights the need for a study targeting the technology's potential and challenges to identify the real and successful application for the technology in the maritime sector for it to achieve its full efficacy and avoid being substandard.

**Table 1.** Blockchain projects' use cases in the maritime sector.

| Usecase | Project | Owner/Proposer | Blockchain Used | Documentation |
|---|---|---|---|---|
| Fuel quality and traceability | BunkerTrace | Blockchain labs for open collaborative and Main Blockchain Labs | Ethereum | https://bunkertrace.co/ |
| Shipment Tracking | TradeLens | Maersk and IBM | Hyperledger Fabric | https://www.tradelens.com/ |
| | GSBN | Oracle, Microsoft, AntChain and Alibaba Cloud | AntChain | https://www.gsbn.trade/ |
| | Silsal | Abu Dhabi Port | Hyperledger Fabric | https://www.adports.ae/abu-dhabi-ports-collaborating-with-msc-mediterranean-shipping-company-on-international-blockchain-solution-silsal/ |
| | Calista | PSA International, and Global eTrade Services (GeTS) | Not a blockchain but an intensive API delivering end-to-end data (based on blockchain concepts) | https://calistalogistics.com/ |
| Track and Trace hazardous goods | (pilot project) | BLOC and Lloyd's register foundation and rainmaking consortium project | – | https://www.lr.org/en/latest-news/lr-foundation-bloc-establish-maritime-blockchain-lab/ |
| Smart Bill of Lading | CargoX | CargoX | Ethereum | https://cargox.io/ |
| | TradeLens | Maersk and IBM | Hyperledger Fabric | https://www.tradelens.com/ |
| | Bolero's digital trade platform | Bolero | Volton Corda based | https://www.bolero.net/ |
| | Easy Trading Connect | Blue Water Shipping Louis Dreyfus | Ethereum Quorum | https://www.ldc.com/press-releases/louis-dreyfus-company-ing-societe-generale-and-abn-amro-complete-the-first-agricultural-commodity-trade-through-blockchain/ |
| | (proof of concept) | Pacific International Lines ,and PSA International and IBM Singapore | Hyperledger Fabric | https://www.globalpsa.com/psa-pil-and-ibm-conclude-a-successful-blockchain-trial-along-the-southern-trade-corridor-stc/ |
| | WAVEBL | https://wavebl.com/about/ | private blockchain | https://wavebl.com/ |
| | Tokio Marine | Tokio Marine Holdings | Corda | https://www.gtreview.com/news/fintech/insurance-blockchain-alliance-leaves-ibm-hyperledger-for-r3s-corda/ |

**Table 1.** *Cont.*

| Usecase | Project | Owner/Proposer | Blockchain Used | Documentation |
|---|---|---|---|---|
| Digitization | (storage on blockchain) | DNV GL | Vechain Thor | https://www.dnv.com/assurance/certificates-in-the-blockchain.html |
| Smart Contracts | Blockconnect | 300Cubits | Ethereum | https://www.300cubits.tech/ |
| | ShipChain | ShipChain | Ethereum | https://www.freightwaves.com/news/logistics-provider-shipchain-which-built-on-blockchain-shutting-down-after-big-payment-to-sec |
| Insurance and finances | (proof of concept: maritime insurance platform) | A.P.Møller-Maersk, Willis Towers Waston, MS Amilin, and XL Catlin | Corda | https://www.wtwco.com/en-GB/news/2018/06/willis-towers-watson-at-the-forefront-of-blockchain-technology |
| | (proof of concept: Blockchain Insurance Industry Initiative ) | B3i Services AG | Corda | https://b3i.tech/ |
| | (enterprise-level blockchain consortium) | RiskStream Collaborativ | Corda | https://web.theinstitutes.org/institutes-riskstream-collaborative-launches-canopy-risk-management-and-insurance-industrys-first |
| | Shipowners | Shipowner.io | Ethereum | https://shipowner.io/ |
| | Skuchain | Founded by Srinivasan Sriram | Hyperledger Fabric | https://www.skuchain.com/ |
| | Provenance | Owned by Morgan McKenney | Ethereum | https://www.provenance.io/ |
| | Tallysticks | Co-founded by Kush Patel | | https://www.f6s.com/tallysticks |
| VGM Portal | SOLASVGM | Kuehne + Nagel Group | Hyperledger Fabric | https://newsroom.kuehne-nagel.com/kuehne--nagel-deploys-blockchain-technology-for-vgm-portal/ |

*3.4. An Extended Spectrum of Use Cases*

A crucial step in the application of blockchain in the maritime sector is the identification of the full spectrum of use cases throughout the sector. Unlocking and pinpointing the technology's complete potential promotes widespread adoption, because despite early adopters and current blockchain projects emerging in the sector, we notice that fragmentation remains as initiatives remain separated and fragmented across distinct actors in distinct use cases.

For this, we refer to not only blockchain characteristics that can allow a widespread complete collaboration across the maritime ecosystem but also that the maritime sector is well aware of the negative effects of fragmentation and redundancy. An example of this awareness is the effort being made to deploy a functional European maritime single window system, based on a "tell us once" principle [31].

Therefore, the following section aims to broaden the spectrum of potential applications and use cases for blockchain beyond the aforementioned scalable decentralized communication system and introduce blockchain for increased efficiency, transparency, and traceability. Additionally, we identified two approaches for implementation.

### 3.4.1. Projection and Pioneered Projection

The simplest way to successfully implement the technology is through projection, where no new hypothesis for functionality is made. The use case is implemented elsewhere, but can be streamlined through blockchain.

To elaborate on this approach, we refer first to the enforced shipping standards by the International Safety Management (ISM) code and International Organization for Standards (ISO). For example, the regulatory quality and safety standards in the shipping sector are anticipated to become more diligent and stricter after the trade union's economic disintegration of primal powers [2], and consequently rendering operational processing increasingly intricate.

Second, we accentuate the fact that various trade goods require specific conditions to project upon the use of blockchain. It simplifies documentation and processing, and also ensures the maintenance of convenient and safe conditions throughout the whole journey of the cargo through the supply chain. An example of such a use case is vaccines. At the beginning of the pandemic, it made sense to utilize air shipping for faster transportation, but as the virus become less virulent to the world population and its spread slowed down, it was worth thinking about a more economical solution such as the use of the shipping industry [32]. However, an economical solution should not affect efficiency, since according to the World Health Organization, vaccines are highly susceptible to degradation when exposed to temperature fluctuations. Therefore, a projection use case is the use of blockchain monitored containers where the temperature is constantly maintained using a passive cooling system and any fluctuation is recorded in real-time. This allows immediate intervention consequently limiting degradation. Smart containers equipped with sensors are currently used in air transport, thus the projection and ledger monitoring can be deemed achievable and successful where documents can be digitized, certified and verifiable [33].

Furthermore, projection approaches can then be pioneered as extended experimental approaches for improved performance. For example, in this case, the use of the internet of things (IoT) can be suggested for complete traceability. Currently, IoT devices are successfully used in the shipping industry, but their use case is restricted to real-time GPS tracking for cargo and ships. However, alone, IoT devices are susceptible to security threats with limited computational capabilities [34]. Thus, the use of blockchain can be justified and encouraged, as the ledger not only monitors goods but also devices as well. Hence, it becomes easier to detect malfunctioning or malicious devices.

Better transparency and traceability reflect operational efficiency throughout the whole process, decreasing costs and time delays as a result [34].

### 3.4.2. Parallelism

As a result of the lack of significant studies on blockchain in the maritime sector, particularly in the domains listed below in Table 2, we highlight another possible approach to identify use cases through our triangulation method. We based this approach first on triangulation, where we inspect and analyze blockchain use cases, literature, and projects in other sectors. Then, we can parallel studies from those sectors into the shipping industry. However, the reasoning at the beginning of our study that initiated blockchain's potential in the maritime sector is of major importance, where we started with the correlation between technological advancement using blockchain and profit growth. This first step captivates the maritime actors' attention in this competitive gain-driven environment. Any innovative cost should have an exceedingly reverse effect on the overall cost. Afterwards, we can elaborate on the use of this technology in other applicable domains inside the maritime sector and use parallelism, as shown in Table 2.

**Table 2.** Extending blockchain's spectrum of domain applications.

| Domain | Brief Elaboration |
|---|---|
| Security | Ref. [35]: Connected systems in the maritime sector can be hacked and [36]: Security attacks have immense consequences. Distributed approaches are more secure than centralized ones Ref. [37], hence blockchain can be deployed for adjoined security features for existing systems such as connected devices [38]. |
| Environmental sobriety | Refs. [39,40]: Disastrous incidents in the maritime sector have led to stern rules. Refs. [41,42]: Rules and standards that are enforced can be largely preventive and beneficial. Ref. [43]: Currently, the enforcement of these rules remains far from an optimum where volitional infringements in 2018 are up to 30%. |
| Counterfeit and malicious activity | Refs. [8,44–55] demonstrate in literature the use of blockchain for fraud prevention and detection. The maritime sector has active concerns where counterfeit and malicious activities raise costs by at least 10% [37] and can immensely benefit from the use of technology having anti-fraud characteristics. |
| Reducing delays and unnecessary third parties | We reflect on the several potentials to highlight the consequent improvement in operational efficiency that aligns with our hypothesis. The prevention and easy identification of malicious activities, the simplified regulatory enforcement, and the adjoined security privileges combined with incremented transparency and traceability and the elimination of unnecessary third parties would unlock colossal benefits for the maritime sector. |

For the technology to reach its full capabilities and revolutionize the maritime sector, it needs not only to be applied in different use cases but also be deployed with a sense of homogeneity. These different applications should be able to interoperate to facilitate secured widespread digitization that propagates through the whole supply chain. Since the technology is still new to this sector, with limited successful applications, it is necessary that we also explore its limitations. We can also use triangulation, projection, and parallelism to anticipate beforehand some of the challenges and try to solve them.

**4. Challenges to Overcome**

Blockchain is still often described in the literature as a novel technology [56]. The term novel represents the recency degree of the technology. The more recent the technology is, the more it is important to dissect it and explore not only its potential but also to understand the challenges it may face. Once challenges are known, the technology can be introduced to the ecosystem levitating the ambiguity it faces. However, the full potential of a decentralized technology such as the blockchain cannot be exploited without efforts being made in the ecosystem to ensure total coordination and widespread adoption. Therefore, we re-address some of the challenges in [1], and bring forth additional challenges with a deeper analysis using our triangulation approach to confirm hypothetical challenges and probe for solutions and analyze them. Hence, the deployment of blockchain technology will become less ambiguous as solutions are identified, listed, and analyzed for suitability.

*4.1. Blockchain Currently Has Limited Use Outside of the Private Sector*

Despite the large variety of pilot projects and described applications in literature, there have not been any concrete blockchain applications in the public sector. Entities of the public sector recognize the massive potential of blockchain but are deterred from participating in this phase where there are still ambiguities surrounding this technology that are yet to be solved. Therefore, this challenge can be linked with other challenges that are developed below such as security threats, network limitations, lack of regulations, and unwanted scrutiny [1].

*4.2. Blockchain Predates Regulations*

Modern technologies always come with ambiguity in the regulatory area. Currently, there is an absence of regulatory oversight around blockchain. Firms and industries are usually reluctant to deal with the uncertainty that comes with recent technologies that pose considerable risks and threats.

In the maritime sector, regulations have sometimes been controversial even prior to the introduction of the ambiguity surrounding blockchain laws. This can be even more complex with the presence of a large number of actors that needs to protect their interests with ever-increasing profit-oriented competitiveness. Therefore, from the beginning of global shipping, efforts have been made to enforce regulations and control. The IMO was founded in 1948 to regulate international shipping [57]. However, the different types of regulations (economic, commercial, environmental, etc.), according to local and international laws, have incremented the dilemma's complexity, despite the IMO's considerable efforts.

Ports slowly adopted regular technological advancement as support to enforce regulations. Generally, every port has a PCS that is used by port officers and others for enforcing regulations locally, reigning over actors across that port, such as terminals and ships, amongst others. This way, regulations are enforced across the supply chain through a series of globally spread PCSs. However, there is little to no communication across these PCSs, and there has been little to no coverage in literature for an envisioned global PCS to harmonize the maritime sector, including data sharing, information exchange, and regulations compliance. The use of blockchain aligns with the regulatory concepts of the maritime industry. It assures the transparency and traceability required and facilitates collective efforts in order to achieve sector-wide consistency. We can envision it as a decentralised distributed infrastructure for the pre-existing PCSs, and used technologies to increase efficiency and communication.

However, up until late 2018, the European Commission (EC) did not regard cryptocurrencies as being real money, without any regulations except opinions and warnings, particularly about price volatility. Despite stating the great potentials of decentralized ledgers and blockchain and launching several pilot projects to distinguish true opportunities behind blockchain from the hype, any amended regulations such as the forth anti money-laundering directive considered only exchanges between crypto and fiat currencies [58]. Thus, until now, no considerable regulation has been made regarding blockchain or cryptocurrencies worldwide. This presents a challenge for the use of blockchain in the maritime sector, considering that some solutions may be built on top of pre-existing crypto blockchains, notably Ethereum, with its smart contract-implementation features. After the bull run that the crypto market has experienced since the beginning of 2020 and the immense increase in investments, governments started acknowledging the need to introduce crypto regulations, and some have. Unfortunately, these new regulations were not always supportive. While El Salvador announced its new law declaring bitcoin as a legal tender, China, the home of most bitcoin miners, took the opposite road and started a mining eradication. France also has started to define regulations considering the most reputable cryptos on the market such as bitcoin, Ethereum, XMR, and so on. The European Union (EU) has also launched a crypto regulatory project, leaked to the press last September, with the intention to harmonize regulations and avoid competition across its states. Different attitudes toward cryptocurrency and the absence of a worldwide regulation project present a considerable challenge for the application of blockchain in all sectors. Even when we succeed in defining these regulations, enforcing them in a decentralized widespread infrastructure without affecting the innovatory aspects of blockchain will become the new challenge.

Moreover, even without the cryptocurrency's conundrum, building a blockchain-based application must comply with pre-existing regulations such as the EU general data protection regulation (GDPR), especially in the maritime sector where data is considered sensitive. After reading the GDPR, we can quickly detect the conflict that blockchain presents. The first blockchain is immutable, while GDPR reserves the right to rectify and

erase data. Second, its decentralization refrains from establishing clear responsibilities and accountability. Finally, lawfulness can only be ensured by node basis, which seems excessive. The literature proposes three approaches to address this challenge. The first is the use of a central authority to enforce GDPR regulations and assume responsibility, as for data rectifying, it can be through a redactable blockchain or each node removing the data and re-calculating all consequent blocks. The second is through distributed responsibility. This approach is similar to the first one; responsibility is assumed through the collective efforts of multiple central authorities. The third is pseudonymization. This can be achieved through data encryption, hashing, and pseudo-identifiers. The third approach was used in a successful pilot project from the German federal office for migration and refugees [59].

The business sector, particularly the shipping industry, would be more willing to apply blockchain once this two-fold challenge has been overcome [60] in addition to the added sense of security once regulations and responsibilities are defined. This will make entities more willing to trust technologies, since misuse is condemned.

### 4.3. Scrutiny: The Result of Transparency

In the previous section, we have detailed the benefits of the added transparency that results in more data sharing, facilitates origin tracking, and eliminates ambiguity across the supply chain. However, this blockchain-added feature can be described as a double-edged sword. It is also the same transparency that might discourage entities from supporting blockchain solutions. The information available to consumers, auditors, or even competitors does not seem appealing to businesses in the shipping industry in particular and to global businesses in general. From an economically unbiased point of view, increasing transparency and traceability results in increased efficiency and accountability. However, the complete adoption of this technology needs actors' support, which we can clearly say has not been complete until now, and this is why the literature coverage of total traceability using blockchain-based solutions remains conceptual [1].

### 4.4. Security Challenges

In spite of its secure reputation and a highlighted strength compared to centralized databases, fraud and security threats remain. In theory, blockchain is described as an append-only immutable ledger. This means that any data recorded on the ledger cannot be changed, and is a point of strength for the technology. However, the blockchain does not ensure that data is not changed before being appended to the ledger. For example, if a blockchain-based application recorded sensors entry, and the sensors have been compromised, the wrong data will be recorded on the immutable ledger [38].

Blockchain systems such as bitcoin and Ethereum operate using the proof of work consensus; despite it being a secure concept, data is not as immutable as theoretically described. The power of immutability remains as long as 51% of the network is not conspiring. The 51% does not refer to network participants but to computational ability. This means any entity that gains enough computational power can hijack the network. This 51% consensus-based protocol puts networks at risk, particularly small ones. A study performed in 2013 on the largest network at the time, Bitcoin, revealed that the consensus is not incentive-compatible and miners colluding can obtain more than individual shares, encouraging selfish mining [61]. Moreover, the secured protocol is not infallible, as the network can be forked [62].

Additionally, public blockchain can be susceptible to denial of service (DoS), eclipse attacks, man in the middle (MitM), and signatures can be deciphered with technological advancements such as quantum computing compromising blockchain data, since it is immutable.

Cryptocurrencies have been the subject of hacking attacks exploiting vulnerabilities in apps, software, protocols, smart contracts, and other points of failure where considerable amounts of money were stolen [63]. These previously exploited points of failure were amended. However, this condemned the previously hailed unhackable technology. Fur-

thermore, double spending is possible with an entity possessing a high percentage of the network's hashing power. This can be catastrophic in the maritime sector, where it takes one document falsification to allow the transport or entry of illegal or dangerous goods.

This is a considerable security challenge for the technology to overcome, and as with any security system, as more vulnerabilities appeared, more complex solutions were proposed to patch them. Therefore, any blockchain-based application used in the maritime sector should be first deployed with a keen sense of security aspects and second maintained through time. Why not combine the security of pre-existing systems with blockchain? We see blockchain as a technology to complete and which does not supersede existing systems. One of the solutions that is currently applied in the maritime sector is the use of private or permissioned blockchain to avoid relying on the security aspects of public consensus protocols. Private and permissioned blockchains have control over participants of the network, which makes enterprises more willing to share information and data and lessen the impact of dubiety with a more restricted ecosystem.

### 4.5. Network and Technical Challenges

The previously mentioned challenges are common to the global adoption of new technological advancement, and often after detecting them adopters will deploy solutions to overcome and flatten the fear of the adoption curve over time resulting in widespread adoption. However, this cannot be said about network or technical challenges. They are more omnipresent.

### 4.5.1. Distinct Technical Capabilities

Currently, enterprises around the globe do not all have the required technical talent needed to adopt blockchain, and enterprises' technical capabilities vary. An initial blockchain deployment in an enterprise will result in initial expensive costs for teams' technical training and equipment. Moreover, the distributed decentralized nature of this technology requires a considerable number of participants for any deployment to be viable. Consequently, this will also increase costs and add to the complexity because it means incrementing the number of adopters that have various backgrounds and are not necessarily familiar with high-technical advancement or can afford equipment costs. This has been pointed out in a study on blockchain adoption in supply chains [29]. The use of blockchain across a supply chain will mean that some users that have limited technical knowledge will have to record data on a high-tech ledger such as blockchain. For example, if we use blockchain for orange traceability, this means that maybe at some point in the supply chain, farmers, who are not familiar with blockchain, will have to record data on the ledger about the orange, such as the time of picking to ensure the good quality and livability of the fruit. Additionally, wrong data or human error cannot be tolerated on an immutable ledger. Meanwhile, the considerably large number of users will make it harder to verify that every data recorded on the ledger is correct. The enormous impact of human error was demonstrated in 2016 when an error in Ethereum's protocol resulted in the heist of USD 55 million [64], which also highlighted the bad reputation of blockchain and its use by dark users in money laundering and illegal activities.

### 4.5.2. Data Storage

Additionally, concerns about storage are often raised in blockchain applications because blockchain is not conceived as recording bulks of data. In the maritime sector, and particularly applications including IoT technologies continually result in considerable amounts of information being processed and permanently stored. We previously discussed the added sense of security that blockchain can enforce in IoT-blockchain-based applications. However, this combination also faces challenges that need to be assessed prior to deployment. Envisioning a global scale solution where each node has a copy of the ledger will result in too much redundancy at some point. This issue has been addressed in the literature using several approaches. The first intuitive approach was to limit data redun-

dancy by using distributed storage. When a block is created, it is fractioned and then fed into a coding algorithm that will further divide these fractions of a block into sub-fractions to be distributed and stored on the network. The blockchain remains distributed but will be collectively stored limiting redundancy. However, this approach might be susceptible to information inconsistency and limits our system's efficiency [65]. Other studies propose compromised approaches between the simplified payment verification method (SPV) and full-node pruning in order to save space, either by deleting expired transactions and keeping only block headers or by introducing the idea of block summary, respectively. A block summary means using a certain algorithm in order to summarize relevant data and delete others. However, both these methods are targeted only to case-specific uses such as financial transactions or land agreements, and cannot be used in cases where data is sensitive to deletion such as identity management or the maritime sector. It compromises the system's transparency and full historical traceability [66,67]. Several studies discussed the use of IPFS for improving storage efficiency. Coupling such technology with blockchain improves the storage system's potential without traceability compromise. IPFS can be described as a content-addressed storage model. A study in 2018 made on the blockchain network proposed that instead of recording transactions in the block, transactions are stored in IPFS and the resulting hash is then stored in the block and used in the Merkle root and the block's resulting hash. When a miner successfully finds the correct nonce, he can broadcast the block for others to verify. Since transactions are usually broadcasted on the network, verifiers would not need to query the IPFS network for every transaction. Transactions in the local mining pools differ usually because of network delays. Thus, other miners will have in their mining pool most of the transactions, apart from a few missing ones that they can obtain by querying the IPFS network through the IPFS hashes without affecting the system's efficiency by a large number of requests. Once the block and its transaction are validated it can be appended to the ledger. The compression ratio is inversely proportional to the data recorded on the ledger. The more the blockchain expands in size, the more the compression ratio decreases. Small transaction data resulted in a large compression rate (>1) in 2009. However, the bitcoin blockchain considerably expanded in size over time; therefore, in 2018, the compression ratio reached an optimum of 0.0817. Furthermore, in theory, optimized storage will increase new node synchronization speeds, which will also have a positive effect on the overall efficiency of the network. This approach is yet to be tested in genuine network situations [68].

Similarly, Nizamuddin et al. [69] detailed the use of the smart contract feature of blockchain, specifically Ethereum, on top of the IPFS network. However, some problems remained: duplication, content piracy, and information availability. When a user downloads a file, it can be duplicated, and another user can claim ownership, and any offline changes to the file are not recorded, which threatens authenticity.

As duplication prevention, multiple approaches were suggested in the literature, such as SPROV [70] which uses encryption and signatures to protect authenticity. However, this method lacks data querying, which is important, specifically in the maritime sector where data is abundant. Other approaches suggested coupling cloud storage with blockchain, implementing real-time hooks for provenance events to be recorded into the ProvChain [71].

Nonetheless, data integrity remained threatened, since all the aforementioned approaches were vulnerable to piracy. Thus, the suggestion of Khatal, Rane, and Patel of an intermediary application between the user, the IPFS network, and the blockchain. The application represents a protection layer. The content of the file is encrypted and can only be decrypted inside the application without being replicated into the users operating systems. Each user is identified by its own smart contract on the network containing the user's metadata (keys, shared files). Once a user is authenticated, he can log into the application using his set of keys. To verify his identity, his smart contract will be fetched through his public key. Then the encrypted registration key on the smart contract can be decrypted, also using his key. If the provided registration key matches the recorded key, then the user is verified and granted access to the application. A user can create a file through the

application editor. The file will be encrypted, and a random secret key will be generated and stored in the owner's wallet. The encrypted file will be shared on the IPFS network. Every deployed file on the IPFS will create a smart contract containing the file's metadata on the blockchain, such as the IPFS hash and the owner's public key. Files on the application can have public or restricted access. In a restricted access mode, the owner can share the file with users through their public keys. Once the creator specifies a user's public key for a file to be shared, a resulting encryption key will be calculated using the public key and the secret encryption key of the file. The resulting encryption key can only be decrypted through the user's (of whom the creator wants to share the file) private key. If a file is public, the creator's public key will be available for all users on the application. The file's smart contract will register the receiving user in both access modes [72].

Nevertheless, these methods discard data availability problems. Despite the improved storage efficiency, one of the biggest IPFS limitations is continuous availability. Node on the IPFS network contain in their cache files they have requested, but if all these nodes go offline, the file will become unavailable. This limitation can be overcome by data replication which is considered a storage problem in the aforementioned approaches, or by creating incentives to keep the data available. The filecoin blockchain provides an incentive layer on top of the IPFS network to guarantee data availability [73].

Furthermore, similarly to Sia [74] and Storj [75], a study suggests the use of erasure coding to ensure data availability. The study introduces an environment based on their concept. The study system's architecture suggests splitting users into two categories based on their needs as well as splitting data into two categories based on its usage. Service providers cannot risk their customers using IPFS storage. Therefore, it is important to define an optimal usage strategy for reassurance. The study was inspired by Blockstack, a blockchain based on the bitcoin network (considered the most secure blockchain network). Adding an additional layer that adjoins the missing features in bitcoin such as smart contracts, privacy functionalities, and traffic handling of decentralized applications. It suggests splitting the network based on the type of users. Service providers are more concerned with data availability, as opposed to regular users. This means they need to take part in the network and maintain a minimum of nodes for protection and reassurance. Thus, the network is split into two parts: one for service providers and the other for regular users. Users can join the service provider's network and one of its nodes as a proxy node, and consequently they will not be responsible for any additional costs for node maintenance or high throughput. Regular users will only need a client with basic data-related features such as uploads and downloads. However, users can choose the profile they want, and regular users have the option to maintain nodes. Similarly to Blockstack, the study chose a layered approach but with tweaks to the layers to enhance the IPFS features. The first layer is the blockchain layer, which means building your own blockchain or choosing an existing one as your infrastructure. Bitcoin was chosen as the infrastructure. The second layer contained the functionalities of the proposed scheme, including transaction verifications. Transactions are verified by the recipient through asymmetric encryption. When a transaction is sent, the sender signs it with his private key, and the receiver can verify the authenticity of the signature with the sender's public key. Moreover, in the study, the keys were linked to the IP of the account and a request functionality is added that allows the node to declare the files in its account. Files in the storage system have two types: mutable and immutable. The third layer is based on the second layer and contains information about the network, such as addresses of accounts and files in the account. Finally, the fourth and final layer is dedicated to data storage. In addition to the immutability option for service providers, they need a more complex strategy to ensure reliability. In order to accomplish this, the study proposes categorizing data as either "hot" for frequently used data or "cold" for far less frequently accessed data. Since the hot data has a higher availability need, it is stored and replicated twice through the network as opposed to using an erasure scheme for the storage of cold data resulting in high data reliability with optimal storage space usage. Of course, both types of data are stored on the IPFS network. As opposed to IPFS with constant high

throughput, the scheme allowed an optimal throughput that only rose when interacting with content service providers, and users can interact with nodes through service providers limiting local interaction and consequently, the constant high throughput [76].

Another study also discouraged directly storing data on the blockchain, but instead of IPFS, it proposed another approach specifically related to the use of IoT technology. IoT has very promising potential in product traceability and quality reporting. However, the industry lacks a standardized data-sharing service. As previously mentioned, the usage of IoT produces bulks of data that can affect the efficiency of the system, both storage- and network-wise, in addition to privacy concerns when recording data on the blockchain. Therefore, the study analyzed the feasibility of a compressed and private data-sharing framework. It used the permissioned Hyperledger Fabric blockchain to track products through industrial supply chains combined with IoT technology and proposed a novel storage architecture. The need for such a framework considerably rose after the EU parliament required actors to provide information that ensures food traceability. The IoT technology also stood out in studies for fraud detection and in the drug industry [77,78].

Nevertheless, in this industry numerous actors adopt various information management strategies and platforms, which makes it hard to ensure traceability with IoT technology on its own. It is also where we encourage the combination of different techniques, such as blockchain for added security. Moreover, in addition to the aforementioned studies, several others emerged suggesting blockchain as a trusted distributed environment for information sharing and storage [79], proving that the use of blockchain and IoT with off-chain compression and encryption strategies in addition to on-chain storage can elevate the system's efficiency and overcome efficiency and privacy barriers. Furthermore, data can be accessed in both modes (off-chain/on-chain) via two distinct types of transactions: point and data. First, the mechanism makes use of pre-existing data flows, and records are internally passed along, arriving at a terminal to be compressed. This particular methodology allows a maximum set of participants to record data through a single compression operation lessening expense. Through internal participants, transactions of a "point" nature are made at each step. Once data reaches the terminal and is compressed, a transaction "data" is made to the ledger. This mechanism addresses the efficiency barrier of recording large bulks of data on the blockchain. Second, data can also be encrypted and only available to authorized parties addressing the privacy issue of blockchain. Data access is managed through an authority that provides keys with corresponding access policies to parties. Two access modes, thus, co-exist through an attribute and policy-based cypher text. Furthermore, industrial parties can be managed on an attribute basis (ABE) and third parties can be managed on an identity basis. Such studies implemented the aforementioned system on the Hyperledger Fabric blockchain, shared across several machines to imitate a true blockchain environment. The study aimed to address the issues in basic blockchain-based IoT designs, where each client has to download the full database with an ever-increasing ledger, and data are usually fragmented across blocks in addition to potential risk threats across the whole chain of participants since they all have open access to the system. Through their proposed mechanism, the data is encrypted and compressed at the end of the off-chain, and uses an ID for each product with the point and data transactions. When a product is passed through actors, each actor will declare a point transaction with the product ID and transfer the product's data to the next until it reaches the terminal. The terminal is a specific entity, responsible at the end for data compression and encryption and accessing policy specification before appending it to the ledger. In order to access the data, a user will query the blockchain for a point transaction regarding an actor's data about the product. The point transaction will point the user to the right off-chain participant. Thus, the user will ask the actor for the data with the product's ID to receive a cyphered reply with corresponding access rights. Now, the user can decompress and decrypt the data. Compared to basic designs, full nodes will only store compressed data increasing storage efficiency. Furthermore, product data is stored whole instead of being fragmented, increasing network efficiency as the user will only need to query a single location for the

needed info. Moreover, data is protected through encryption and various types of access policies, such as ABE, reducing key overuse.

Even though this approach resolves the efficiency and privacy problems of combining IoT with blockchain, some issues remain, such as key fault tolerance and time constraint data. To address the first issue, the study suggests key sharding [80]. In addition to shredding, fully homomorphic encryption, to ensure full protection, can also be suggested [81]. The second issue is that certain data becomes useless over time, and storage can be saved. Therefore, the possibility of data redaction is proposed via a key managed by the access control authority. Additionally, unrelated block offloading filters can be used to store cold data in caches [82]. Furthermore, a suggestion to avoid bottlenecks is sorting relevant information; not all shared data might be relevant or used for upcoming actions. The study shows prominent results regarding feasibility. Experimentation resulted in an increase up to nine times in storage efficiency, and up to an increase ranging from five to twenty times in data access with great scalability potentials. Additionally, other studies that suggest improvements such as:

- A credit-based proof of work with a directed acyclic block architecture [83];
- A light chain concept with a more environmentally aware consensus mechanism [82];
- A new multi-center architecture to enhance privacy and security [84];
- Recycling an already-deployed smart contract for space reduction [85] that aligns perfectly with the compressed and private data-sharing concept and can be introduced as improvements.

### 4.6. The Nexus between Code and Logic in Business

Recent innovative technologies often come with a large gap between technical capabilities and a lack of knowledge of sectorial logistics. Most enterprises, even those willing to take part in the blockchain adoption and who have coding experts, often lack expertise in the blockchain domain and cannot set clear expectations for blockchain-based applications. The hype around blockchain and the success stories of the technology encourages diving into it sometimes without a thorough study of the application, which can result in failure. The use of technology should not only aim to solve a problem, but it should also be the optimal way to do so, thus identifying the correct use opportunity. In spite of the general ambiance of interest in blockchain and acknowledgment of its power to restructure business as we know it, particularly in the maritime sector, the number of applications that identify the specific use of blockchain is still very low. Moreover, there is often a need for an optimal understanding of the business logic in order to identify the right application and deploy the correct codes.

### 4.7. Smart Contracts

Codes on blockchain include smart contracts, an automated self-executing code that runs on the blockchain once its pre-determined requirements are met. Prior to the blockchain, such contracts were not possible; the enforcement of contracts usually required third parties. The emergence of smart contracts led to a new improved generation of blockchain, where contracts can automatically self-execute in a decentralized environment.

A smart contract is a code stored on the blockchain that users can use by sending transactions to its specific address. The enforcement of such contracts relies on blockchain technology and its consensus protocol. This ensures more efficient operations in terms of speed, costs, and transparency. The most known blockchain for its smart contract feature is the Ethereum blockchain. The Ethereum virtually distributed state machine (EVM), spread across nodes, is responsible for running the contract code. The contract code is written in a programming language such as Solidity and compiled into bytecodes to be distributed on the EVM. When someone calls the smart contract, the EVM machines run the bytecode, calculate the result, and update the blockchain's state if needed.

Despite the smart contract's revolutionary aspects, it also has several limitations. For one thing, the execution of a smart contract relies on the existence of the required fund-

ing on the blockchain. This means that any transaction involving cryptos is only executed if the crypto is native to the blockchain and the amount exists on the blockchain. Therefore, transactions cannot be enforced in the absence of these two conditions. A third regulatory party can act as an enforcement authority, but this negates the decentralized authority concept of blockchain. Alas, stakeholders are susceptible to not being paid. For another example, most used types of transactions in the maritime sector are based on fiat currencies that blockchain cannot enforce, but only record. According to Greenspan [86] smart contracts and fiat currencies cannot mix. This means that blockchain applications in the maritime sector cannot guarantee crypto payments and cannot enforce fiat-based payments.

Tests have been made to explore the use of cryptos and tokens in shipment payments, but actors had a hard time relating to the use of currencies other than fiat, that can also be used outside the platform and are relatively stable, compared to cryptos such as 300Cubits. Greenspan also highlights the nexus between code and business logic where an unclear notion of decentralized code can result in overblown expectations in addition to a waste of valuable resources on unachievable ideas.

### 4.7.1. Blockchain Is Isolated from the Real World

Similarly to fiat currencies, interactions with blockchain such as external services are also not possible. Blockchains are a protected decentralized isolated environment from centralized real-world data exchanges. Blockchain has no access to real-world data, therefore smart contracts involving real-world resources cannot be enforced [38]. This means that any interaction with the blockchain not only negates the concept of blockchain but is also not possible due to the blockchain's core concept that any recalculation of transactions on the blockchain at any given time should always result in a collective consensus of the blockchain's state across nodes. It is the deterministic nature of this technology. Real-world data, such as API requests, are always changing, which means they are not allowed on the blockchain. However, oracles can act as a middleman to successfully link blockchains with the real world. They provide reliable data from scopes that blockchain cannot access since a blockchain has no access outside its data, such as internet-provided data.

Using oracles, we can code smart contracts to rely on data from the real world, such as temperatures. They are not specific devices, but more of a concept, similar to the internet; everyone knows what the internet is, but it is not a single device. Therefore, any source of information that feeds data to the blockchain from the real world is an oracle. They do not store data on the blockchain, they collect data and when a smart contract is executed that requires information from the real world, the code collects the needed data from the trusted oracle, such as an IoT device, as seen in Figure 6.

Oracles can be software devices, hardware devices, humans with specialized knowledge, or computational devices. Software oracles can access online information from any source of data. Hardware oracles are usually devices that have access to real-world data, such as barcode scanners, IoT or RFID devices, or sensors. Human oracles are skilled and trusted entities who can search for the needed information and authenticate it such as crypto experts or governments. All the previously mentioned oracles have a request-response role. Oracles can also use their computational power to limit on-chain gas usage. They can perform beneficial off-chain calculations. Furthermore, oracles have a two-way communication channel with blockchains. They can provide information for smart contracts but can also provide information to the outside world from a smart contract. In all of the previous data patterns, the blockchain queries the oracle for information to be provided back. There are also additional design pattern oracles such as publish–subscribe and immediate read. A publish–subscribe pattern is useful for changing data. Upon change, the oracle can broadcast the information so the provided information can always be updated with changes happening in the real world. Another type of pattern is the immediate read, where oracles store the needed information so it can be provided immediately upon request without having to query an outer source [87,88].

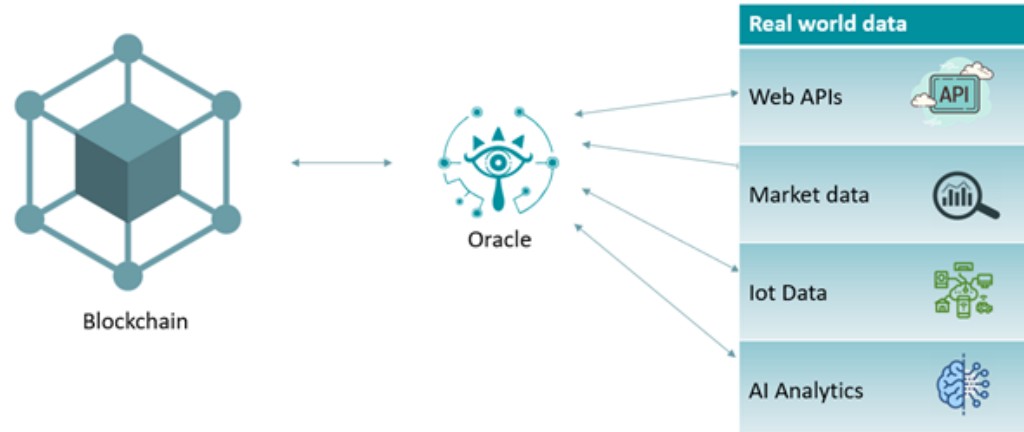

**Figure 6.** Oracle role representation in a blockchain environment elaborated by the author.

However, the obvious problem of trust rises again. Oracles are important for the successful implementation of smart contracts that involve real-world data; however, they also add to the system's complexity, putting its security authentication and trust at risk. Egberts [89] described using an oracle as a "two step-back from decentralization" for two main reasons. Oracles should be trusted sources. Since they are generally not distributed, centralized and trusted oracles bring back the problem of a single point of failure that cannot be tolerated in a blockchain environment. In addition, oracles have non-deterministic data, therefore re-introducing trust issues. Oracle should be trusted, hence, they reintroduce the need for trusting one single entity and as a consequence compromise our peer-to-peer established trust.

Similarly to the previous trust problem, this can be solved by decentralization. Several projects have been launched to implement a decentralized blockchain oracle service. Provable is a decentralized oracle service launched in 2015 and used by several blockchains for smart contracts such as Ethereum, Hyperledger Fabric, Corda, and others. Provable can testify that the returned data from its decentralized chain is authentic by providing authentication proof. This solves the oracle problem. Private and public blockchains can access real-world data through a decentralized trust environment where Provable can testify for data authenticity that it acquires from service providers, without them having to worry about blockchain compatibility. Another decentralized oracle service is Chainlink. Chainlink was originally built on top of the Ethereum blockchain with the intention to be more blockchain scalable in the future. Chainlink provides a continuous feed of multi-sourced live data across the globe to ensure reliability. For any user to be able to provide data on the network he needs to stake the platform token in order to claim rewards. Thus, false information will result in the loss of these tokens and their money. Therefore, data reliability comes from incentivizing users to provide correct information. There are many other decentralized oracles projects that are launched such as Astraea [90], Dia [91], and Band protocol [92], etc.

In conclusion, regarding oracles, when choosing to work with external data sources, the chosen trust model can considerably impact the environment's nature. This is why when working with blockchain an optimum oracle trust model should also be decentralized that can provide trusted data to the blockchain.

4.7.2. Interoperability

We identified several emerging blockchain projects in Table 1 with some of them having similarities and the potential to revolutionize the way things are done. However, one distinction worth mentioning is that these projects run on different types of blockchains. This brings us to the blockchain interoperability problem, as we cannot argue PCS's lack of

interoperability, yet can provide multiple detached blockchain solutions.For that, we circle back to the same fragmentation.

The term interoperability signifies cross-communication between these several types of blockchains. This can also be referred to as the oracle paradigm, but instead of isolation from an external source, it prevents cross-blockchain communication. Evidently, blockchain projects should be implemented with interoperability in mind. Ref. [60] highlighted a high level of awareness of the benefits of blockchain adoption even in developing countries, whereas [93] showed more feeble technological advancements. This put forward a reigning sense of willingness to lift the global digitalization levels for the industry with blockchain in mind, leveraging its capabilities (security and immutability). However, natively, interoperability is not a blockchain feature since each chain has a different infrastructure with distinguished codes and standards.

At present, base-layer protocol communication is obstructed by the implementation of smart contracts into the blockchain fabric (Ethereum, Cosmos, etc.). Hence, trust is limited to the blockchain itself. Any asset transfer from an outside source raises the question of validity, where each blockchain questions the validity of the other. Consensus is what determines canonical history and condemns validity, which is extremely cumbersome in public blockchains, for example, in depicting the secure nature of these networks. Thus, a base layer communication interoperability signifies that chains should be able to understand and process the history of one another for asset exchange. That can be extremely difficult given the distinctive, various and sometimes cumbersome consensuses; cross-chain interoperability was deemed impossible under its classical definition by [94]. However, the variety of different projects with distinctive functionalities raises the need for an interoperability feature that allows access to the whole plethora. This paradigm has yet to be onboarded in the maritime sector. Ref. [95] discuss interoperability challenges in detail such as efficiency, atomicity, and security. It also describes interoperability schemes with two modes: passive and active. The passive mode corresponds to a blockchain monitoring another. The active corresponds to receiving feedback after sending data to another blockchain. Additionally, it discusses the aforementioned base layer interoperability and proposes establishing it on different layers.

There have also been pioneered interoperability solutions proposed by different projects such as PolkaDot, Cosmos, Cardano, Sidechains, Plasma Bridge, Lisk, and cross-chain interoperability protocol by Chainlink and others [96]. These projects treat the problem from diverse angles, and vary in their current progress level. We refrain from assessing in detail these projects and situating them for the maritime domain. However, we emphasize the necessity for any technological shift in the maritime sector to be interoperable and thus utilize these solutions to achieve it. This is also well highlighted by the future direction of digitization projects in the maritime sector, where interoperability is key, whereas the SW for ship data exchange is to become mandatory. This is according to the amendments to the facilitation convention adopted to enhance digitalization that is expected to enter into force on 1 January 2024 [97]. We detect the interest of the UNCEFACT in blockchain for trade facilitation [98] and possibly its integration with the SW project [99]. Thus, any technological deployment should align with the global efforts made by the OMI and the corresponding local efforts, where, for example, the European SW is expected to streamline the electronic data exchange between customs and non-customs domain through a legally centralized framework. It is noteworthy that the regulation for the European include the international regulations in addition to other European custom regulations which associates coherently with interoperability concepts.

### 4.7.3. Common Code Sense

Smart contracts are codes and codes are susceptible to bugs and hacks. In immutable automated transactions, bugs can have devastating impacts. Therefore, smart contract deployment should be delicately addressed with careful prior validation. A hack in the decentralized autonomous organization (DAO) that exploited a code vulnerability resulted

in the theft of more than USD 50 million worth of Ethereum [100]. That was the cause for Ethereum's hard fork, where its main developers reverted the blockchain state back to before it was hacked, and Ethereum classic, which is still hacked. This enormous hack pointed out the importance of assessing smart contracts' security. Studies have been conducted to help prevent such vulnerabilities prior to deployment detection to ensure correctness and safety.

Since smart contracts have usually access to large sums of coins, this makes it more incentivizing to explore their vulnerabilities. With the increasing number of smart contracts, detecting vulnerabilities is a must. More literature has emerged since 2016 that has focused on the assessment of smart contracts' security. The literature coverage took two paths; the first focused on correct coding and overcoming bugs, and the second consisted of tools dedicated to detecting vulnerabilities in smart contracts.

Correct coding is a great way to prevent security issues. A study conducted based on teaching smart-contract coding exposed several coding mistakes with ways to overcome them. It takes a simple "rock, paper, scissors" smart contract running on Ethereum to demonstrate that even with a simple case, smart contracts are complex. This manifested in Etherpot; an application built on Ethereum for a lottery that was later deemed erroneous [101].

However, even a very skilled programmer is prone to coding errors and undetected bugs; this is why some studies went even further with developing smart contract bug-detection tools, further developed in [1].

### 4.7.4. Unquantifiable Clauses

In addition to technical coding issues, there are also unquantifiable clauses of contracts. These clauses might also lead to exploits, since they cannot be formally coded for a machine to be correctly executed. This can be countered by a recent emerging study. Smart contracts are various and perform different activities, but they have some common specifications, which was the ground on which this study was made. To quantify unquantifiable clauses, a tool should be created to make it easier for users to define a smart contract's specs. A parameter used to test one contract might be different from another, which is why we cannot formally verify contracts. Therefore, the study recognizes that the most important part of verifying the correctness of smart contracts is not the process of verification but knowing its specs and what the program is supposed to do. Thus, specs' formalization is important in the creation of a cross-platform detection tool. Since traditional smart contracts do not have formal specs, they cannot be checked. Therefore, the paper suggests implementing reusable invariants which can be checked. Invariants should be consistently true at any stage of the program. These invariants should be stated formally, so the specs can be cross-platform and the same across various contracts. Their first step was interacting with the community to collect bug detection invariants and prove their usefulness with their possible formal representation. The second is to feed those invariants to a Certora prover [102] which is a tool used along with existing smart contract compilers to detect bugs. The tool was used on several Ethereum contracts. Results countered the unquantifiable clause fault that suggested automation is not feasible. The study concluded that since various smart contracts comply with the same formal rules, such a tool is extremely promising and cost-effective [103]. This last study implies that manual auditing can be overcome, and automation is feasible in addition to specs formalization.

### 4.8. Consensus Protocols

Blockchains are decentralized ledgers that ensure trust through technology. The network uses a consensus mechanism to ensure consistency and data validity across its nodes. However, this consensus does not come without an expensive cost which discourages the technology's implementation in the maritime sector. As well as the derivative fees that come from deploying new technology, having teams trained, and the transaction costs that

are required to write on ledgers, the way the network reaches a consensus state can also be costly.

The most commonly known consensus for blockchain is proof of work (PoW). PoW is currently used on the largest blockchain network: Bitcoin. It was also previously used in Ethereum, before consensus switched to a less energy-intensive consensus, called proof of stake, in 2022. In PoW, transactions are broadcast on the ledgers to be added to the next block that will be added to the ledger. A node can add the block to the network after calculating a random value called the nonce. The nonce, combined with the hash of the block, needs to comply with the network's difficulty level. When a node finds the correct nonce it will add the block including its nonce to the ledger and broadcast the result, obtaining the incentivising reward. Other nodes on the network declare their validation of the newly added block by adopting it to their ledger and including it in their next block calculation. The nonce is part of the cryptographic properties of blockchain; they are hard to calculate but easy to verify. Calculating a nonce is computationally and time-consuming, and generally calculated by trial and error or brute force. Therefore, PoW comes with considerable financial costs. The calculations require a great amount of electrical power. In addition to the constant incentive for developers to update their equipment because with better hardware performance, there is more chance of finding the nonce and therefore more chance of being rewarded.

To understand the large financial and governmental costs of such a consensus, we will detail bitcoin's and Ethereum's consumption indexes. A single bitcoin transaction today has a carbon footprint of 841.07 $KgCO_2$ which is equivalent to watching 140,178 h of YouTube or making 1,864,103 visa transactions. Furthermore, its power consumption is around 1770.67 kWh, which is equivalent to the power consumption, over 4 months, of a typical French household. The annualized total bitcoin carbon footprint of bitcoin in October 2021 is 83.72 $MTCO_2$, equivalent to the country of Bangladesh, and its power consumption is equivalent to Poland, with 176.25 TWh and with an accumulated electronic waste of 24.17 Kt, which is similar to the small IT equipment waste of the Netherlands [104]. A single Ethereum transaction was comparable to a typical French household consumption over 13 days and had a carbon footprint of 84.71 $kgCO_2$, similar to watching 14,118 h of YouTube. The annualized electrical consumption in October 2021 was 79.17 TWh, equivalent to the power consumption of the country Chile and the carbon footprint of Trinidad and Tobago combined with 37.61 $MTCO_2$ [105].

The consumption costs are massive, both financially and environmentally, especially with the omnipresence of increased environmental sobriety. The willingness to adopt the technology is related to those factors. Costs have a significant impact on the success measurement of technology and as long as costs are relatively high, willingness to participate will be relatively low. Ethereum has gradually switched to a proof-of-stake type of consensus that decreases its power consumption.

Ethereum and bitcoin are public blockchains. To avoid considerable costs, it seems more optimal to use private blockchains such as Hyperledger Fabric. This uses a consensus that does not require any special hardware or requirements other than servers already present in normal organizations, and is not power-hungry. This would be effective especially in the maritime sector, where regulations are becoming more and more strict, and green energy is becoming more common, with a general sense of environmental awareness.

## 5. Discussion

The maritime sector has taken into consideration studies' results highlighting the impact of unhindered digitized collaboration over the global operational efficiency, security, and durability [106]. Considerable actions have been set in motion, such as the European maritime single window, announced after adopting the https://eur-lex.europa.eu/legal-content/FR/TXT/PDF/?uri=CELEX:32019R1239 (accessed on 30 January 2023). This aims to introduce standardization for port calls, ensuring a formality in data exchanges for each national maritime single window for better coordination and operations facility. The regu-

lation, declared to be effective starting August 2025, imposes (among other requirements) nationwide data centralization, extended standardization, and continent-wide (European) data harmonization. Before the due date, prominent efforts will be made by the European Commission Services, the member states, the national single windows, and port operators to assure an interoperable architecture.

Apart from the nomenclature analogy between blockchain and supply chain, the technology has interesting similarities between its technical and technological concepts and those targeted by the maritime ecosystem for faster and leaner logistics in global trade. The operational management in the supply chain is onerous and time-consuming, propagating among a tension-aggravated global web of port-based clustered actors.The tension can be directly linked to the lack of transparency generating skepticism and consequently processing redundancy [107]. Faced with such problems, and to keep pace with the technological advancements that revolutionized other sectors, big companies in the shipping sector started experiencing novel technologies that averted from centralized to the novel peer-to-peer network approach. Therefore, this study takes as its fulcrum that indeed blockchain technology proffers characteristics that conform with desirable operational aspects for the sector. It sets a theoretical conceptual hypothesis for a re-imagined maritime blockchain-based ecosystem. This guarantees security and transparency while overcoming the lack of trust conundrum between its actors. The undisrupted replicated ledger ensures integrity through its peer-to-peer verification system. Similarly, [108] agree that blockchain can act as the brain of the supply-chain body, simplifying the real-time operation handling that might involve no less than hundreds of transactions [109] through a series of decisions and exchanges among various entities. The notarization of data allows transparent traceability against fraud.

However, despite its theoretical alignment, hesitancy still reigns over the technology, and actors remain reluctant to allocate needed resources to the technology and allow it to fully demonstrate what it has to offer.The literature argues that the inhibiting factor is the presence of a gap between blockchain and the existing approaches, and that this gap can be reduced by adopting the unified theory of technology acceptance, using it to converge towards widespread use. We agree that there is a gap between the conventional approaches of the maritime domain and the innovative revolutionary methods proposed and endorsed by the use of blockchain. This has been covered in the first section of this paper, which introduced the deep-rooted conventional PCS approach spread across the globe. However, we do not argue for the efficiency of the current approaches but identify possible weaknesses that the sector has also detected, and propose a possible improvement aligning with the sector's effort through blockchain. Therefore, before stating our research rationale, we also started by defining and describing its technical aspects so we can assimilate them into the maritime sector.

The main motive behind this study is to reposition and convey blockchain applications for the sector to tangible widespread applications, shifting our re-imagined ecosystem from theory to demonstration. It is derived from the fact that maritime actors are well aware of the need for digitization and barrier elimination. Yet, hesitancy obstructing full adoption and tangible gains remain reigning across the sector. Despite the partial recognition of the technology's potential by several emerging projects, we do not fully attribute this to the gap described in the literature.As aforementioned, we partially agree that there is a gap present between both the technology and the current approach. Nevertheless, we also argue that this gap is not due to a lack of technological capabilities, but to willingness that portrayed the sector as prudent and conventional. This willingness can be intensified through a full extensive analysis of the technology from a sectorial point of view. We also used a triangulation approach to demonstrate the liaison between literature and tangible projects and applications.

In the second and third sections, we aim to cover the deficiency of complete academic literature which highlights the technological perspectives in the sector. The fragmentation existing in the literature and fragmented projects lack the extensive analysis presented in

the study. We approached the technology's potential from several domains and presented the full spectrum of the technology's use cases. We anchored not only current blockchain applications in the sector, but also used approaches such as projection and parallelism to widen the spectrum. Through projection, we identified use cases that can be streamlined by the use of the technology that is already digitized, and where functionality is not tested but validated. Through parallelism, we benefited from applications in other sectors, readjusting and applying them to various domains of the maritime sector. As an example, we can refer to [110] where distribution and coordination offer optimum dispatching and improved performance. We are aware that the maritime sector is a business cluster. Hence, we first approached the technological introduction from a profitable perspective. We first clarified the dividends of the innovational investment before diving into its technical impactful advantages, such as overcoming data atomization and inaccessibility, real-time traceability, transparency, better production, and transportation management, etc.

Notwithstanding the technology's full spectrum of capabilities, we differentiate between the existing types of blockchain [111]. We argue the compatibility of one type with the range of needs present throughout the whole supply chain. Thus, we put forward the idea of a hybrid network that combines both the trustless platform of the public blockchain and the restrictions of consortiums and private networks. This aligns better with the data-access policies and competitiveness that characterize the heterogenous maritime ecosystem. As for blockchain interoperability, the advancements on the subject have been well highlighted by [96], revealing a broader context of interoperability than a cross-chain asset and crypto exchanges. Moreover, to enrich our contribution, we transcend. Not only do we highlight potentials, but also recognize that the deployment of this technology in the sector goes beyond partial nescience and technological endorsement, and we try to cover the deficiency concerning the maritime sector and challenges and explore ways to overcome them.The technology, as with any other novel subject, has its challenges; some are sector-specific, and some are common with other sectors, particularly business-based ones. Albeit, our challenge's inspection is maritime biased.

The technology faces the same challenge that the maritime sector is currently addressing: heterogeneity. The maritime sector is characterized by its distinct actors, where both private and public actors collaborate. Thus, technological adoption should be homogenous for it to succeed. Currently, despite considerable standardization efforts for both data (such as bills of lading and manifests), and regulations, systems remain fragmented with no viable noticeable blockchain application in the maritime public sector. Moreover, regulations should not only address the ecosystem, but also the technology itself. Being novel, there are still controversies concerning blockchain and smart contracts to be addressed by the law for lucid accountability. Additionally, before blockchain deployment, there is a need to clearly define transparency aspects to be agreed upon by the network's actors, and to determine the types of networks needed for each operational feature. Furthermore, despite the secure reputation of the technology, any deployment should also be addressed with a clear sense of security.

The main aim behind this challenges assessment was to point out that researchers are aware of the shortcoming of the technology and can provide discernible in-depth discussions to avoid any substandard implementation. We do not portray blockchain as the magical solution to all maritime problems. It has limitations, and can be obstructed by certain characteristics of the sector. However, these challenges are well addressed, as proven through a triangulation approach. We bring forward, as an example, the storage challenge, where we successfully identified a series of literary and applicative projects to answer each situation's need. At the end, the deployment of blockchain with the innovatory sequel combination of these several approaches resolves the challenge. Another focal example were oracles, smart contracts, and unquantifiable clauses, to each of which, through enough research and analyses, we presented and proposed solutions. This proves that, regardless of the challenge, with enough resources and willingness a solution can be found. We anchor the benefits of digitization in general and highlight the use of blockchain as a digital

solution, in particular, where the deployment can have a substantial revolutionary effect on the shipping industry.

At last, we quote the commission official responsible for the European maritime single window: "It is above all a question of harmonizing messages between the various national systems in a simple way." It is necessary to not over-complicate things. For us, this means not undermining or eliminating existing systems, but working with them. We have demonstrated in this article that blockchain can solve the problem of trust in global logistics and that the technical constraints can be addressed. Hence, incorporating blockchain into the framework of SW should be considered. This addresses the urgent requirement of stakeholders in the shipping industry, including maritime authorities and customers, to utilize blockchain technology for greater efficiency, reduced transport costs, and lower energy consumption, by providing end-to-end services. Additionally, increased trust aligns with the decarbonization efforts in the sense that more reliable information exchanged between actors means a better organization because the data is trusted. Future studies can focus on evaluating the effective integration of existing systems and developing widely accepted technological and operational standards for better blockchain integration and interoperability to reach the desired scalable secure communication system.

**Author Contributions:** Conceptualization, R.A., J.B. and C.B.; methodology, R.A.; software, R.A.; validation, J.B., C.B. and C.D.; formal analysis, R.A.; investigation, R.A.; resources, J.B., C.B. and C.D.; data curation, R.A.; writing—original draft preparation, R.A. and J.B.; writing—review and editing, R.A., J.B., C.B., C.D. and F.G.; visualization, R.A.; supervision, J.B., C.B., F.G. and C.D.; project administration, J.B., C.B., C.D. and F.G.; funding acquisition, J.B., C.B. and C.D. All authors have read and agreed to the published version of the manuscript.

**Funding:** This research was funded by Haropa Port and the French "Association Nationale de la Recherche et de la Technologie".

**Institutional Review Board Statement:** Not applicable.

**Informed Consent Statement:** Not applicable.

**Conflicts of Interest:** The authors declare no conflict of interest

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
