# Peer review of "An Extensive Preliminary Blockchain Survey from a Maritime Perspective†"

_smartcities, doi:10.3390/smartcities6020041_

Round 1

Reviewer 1 Report

The paper is comprehensive and presents an attemt to explain very complex problem of wide adoption of Blockchain in maritime.  Maybe, authors should mention in the paper that Tradelens is not in operation since January 2023, and to give some explanations if possible. Also, the question why some participiants in maritime supply chain are not so keen to adopt Blockchain; why they have concerns, or why they are sceptical? Governmental support and legal issues might be touch upon as well.   This is very complex technology, and maritime business and industry are complex, so it is not easy to cover everything especially when the scope is wide like in this paper.

Please check the lines 998 and 999: Ethereum uses PoS not PoW.

Concerning maritime Blockchain integration and interoperability, the following reference should be included:

Kapidani N., Bauk S., Davidson I.E.A., Developing Countries’ Concerns Regarding Blockchain Adoption, J. Mar. Sci. Eng. 2021, 9(12), 1326; https://doi.org/10.3390/jmse9121326

Kapidani N., Bauk S., Davidson I.E., Digitalization in Developing Maritime Business Environments towards Ensuring Sustainability, Sustainability 2020, 12(2), 9235, https://doi.org/10.3390/su12219235

Author Response

Dear reviewer, 

Thank you for your valid and substantial suggestions. 
We have addressed them with care as depicted in the attachment below. 

Best regards, 
Authors. 

Reviewer 2 Report

Dear Authors,

I have a few concerns.

1. Please check the e-mail address of all the authors.

2. The abstract is not clear.

3. Please elaborate more on how blockchain technology reduces the cost. Maybe a table can be provided. Also, a detailed subsection must be added for the cost analysis involved with implementing blockchain technology.

4. The readability of the paper is poor.

5. What makes your paper better than other published papers?

6. line 500,501 --The business sector, particularly the shipping industry would be more willing to apply 500 blockchain once this two-folded challenge has been overcome. Give a numerical or comparative analysis (maybe a table) to prove this.

Author Response

Dear reviewer, 

Thank you for your valid and substantial suggestions. 
We have addressed them with care as depicted in the attachment below. 
And changes to the paper have been done in tracking mode.

Best regards, 
Authors. 

Round 2

Reviewer 2 Report

This paper can be accepted in the present form.